Antitumor activity of Chlorella sorokiniana and Scenedesmus sp. microalgae native of Nuevo León State, México

Reyna-Martinez Raul 1
Gomez-Flores Ricardo 1
López-Chuken Ulrico 2
Quintanilla-Licea Ramiro 1
Caballero-Hernandez Diana 1
Rodríguez-Padilla Cristina 1
Beltrán-Rocha Julio Cesar 2
Tamez-Guerra Patricia patricia.tamezgr@uanl.edu.mx patamez@hotmail.com 1
1 Facultad de Ciencias Biológicas, Universidad Autónoma de Nuevo León , San Nicolás de los Garza , Nuevo León , México
2 Facultad de Ciencias Químicas, Universidad Autónoma de Nuevo León , San Nicolás de los Garza , Nuevo León , México
Silva Pedro
Electronic publication date: 2018 Feb 9
Publication date: 2018
Volume: 6
Electronic Location ID: e4358
Received 2017 Sep 27; Accepted 2018 Jan 22
Copyright: ©2018 Reyna-Martinez et al.
Copyright year: 2018
Copyright holder: Reyna-Martinez et al.
License: This is an open access article distributed under the terms of the Creative Commons Attribution License, which permits unrestricted use, distribution, reproduction and adaptation in any medium and for any purpose provided that it is properly attributed. For attribution, the original author(s), title, publication source (PeerJ) and either DOI or URL of the article must be cited.
License URL: https://creativecommons.org/licenses/by/4.0/

Keywords: Apoptosis, Cytotoxicity, Microalgae, Tumor cell toxicity, Mouse lymphoma, Lymphoproliferation

Funding: National Council for Science and Technology of Mexico (CONACYT) 000496 Laboratorio de Inmunología y Virología (LIV-DEMI-FCB-UANL) CT294-15 This research was supported by the National Council for Science and Technology of Mexico (CONACYT) to Raul Reyna-Martinez (scholarship No. 000496), the Laboratorio de Inmunología y Virología (LIV-DEMI-FCB-UANL) and PAICYT to Patricia Tamez-Guerra (grant No. CT294-15). The funders had no role in study design, data collection and analysis, decision to publish, or preparation of the manuscript.

==============================
Cancer cases result in 13% of all deaths worldwide. Unwanted side effects in patients under conventional treatments have led to the search for beneficial alternative therapies. Microalgae synthesize compounds with known in vitro and in vivo biological activity against different tumor cell lines. Therefore, native microalgae from the State of Nuevo Leon, Mexico may become a potential source of antitumor agents. The aim of the present study was to evaluate the in vitro cytotoxic effect of Nuevo Leon regional Chlorella sorokiniana (Chlorellales: Chlorellaceae) and Scenedesmus sp. (Chlorococcales: Scenedesmaceae). Native microalgae crude organic extracts cytotoxicity against murine L5178Y-R lymphoma cell line and normal lymphocyte proliferation were evaluated using the MTT reduction colorimetric assay. Cell death pathway was analyzed by acridine orange and ethidium bromide staining, DNA degradation in 2% agarose gel electrophoresis and caspases activity. Results indicated significant (p < 0.05) 61.89% ± 3.26% and 74.77% ± 1.84% tumor cytotoxicity by C. sorokiniana and Scenedesmus sp. methanol extracts, respectively, at 500 µg/mL, by the mechanism of apoptosis. This study contributes to Mexican microalgae biodiversity knowledge and their potential as antitumor agent sources.

Introduction

Most commercialized drugs are synthetic derivatives from natural products, or are the result of the systematic screening of terrestrial organisms, such as plants or microorganisms.

Analysis of molecules produced by aquatic organisms has shown that microalgae synthesize a large number of bioactive compounds, including pigments, sterols, polyphenols, fatty acids, proteins, vitamins, alkaloids, and sulfated polysaccharides. This group of microorganisms is extremely diverse and represents a number of unexploited natural sources for bioactive agents. Furthermore, the microalgae intake of polluting elements such as nitrogen, phosphorus, and sulphur for their own growing can be considered an advantage, since such elements can be also metabolized by harmful aquatic weeds to proliferate.

Microalgae are unicellular, simple, primitive, and photosynthetic organisms, producing bioactive compounds for pharmaceutical and biotechnological applications (Shanab et al., 2012; El Baky, El-Baroty & Ibrahim, 2014; Shalaby, 2011), which have shown antiviral, antimicrobial, immunomodulatory, and antitumor properties (Lordan, Ross & Stanton, 2011; Teas & Irhimeh, 2012).

Common failure of conventional therapy against cancer indicates a critical need for beneficial alternative therapeutic agents (Rengarajan et al., 2013). Antitumor activity of microalgal compounds can be explained by their ability to cross the lipophilic membranes and interact with proteins involved in apoptosis. In addition, several microalgal compounds induce DNA-dependent DNA polymerases inhibition, cyclins expression alteration, or major transduction pathways interference. Microalgal compounds have been related to immune response stimulation (Baudelet et al., 2013), as well as cytotoxic against several cancer cell lines (Shanab et al., 2012; Lin et al., 2017). If any compound shows cytotoxic activity against cancer cells, it is important to discriminate if this compound does not represent a threat to normal cells and if its cellular toxicity mechanism is via necrosis or apoptosis. Both apoptosis and necrosis can occur independently, sequentially and/or simultaneously, where the stimuli degree and/or type determines either apoptotic or necrotic death cell (Elmore, 2007). In some cases, cancer chemotherapy treatments result in DNA damage, leading to apoptotic cell death (Elmore, 2007). Apoptosis involves DNA damage and caspases activation. Chlorella spp. extracts have resulted in cell death via DNA damage (Yusof et al., 2010) and caspases activation, demonstrating the apoptosis pathway (Lin et al., 2017).

The aim of the present study was to evaluate the potential of Nuevo Leon, Mexico native microalgae, C. sorokiniana and Scenedesmus sp. extracts, isolated from Nuevo Leon, Mexico, against murine L5178Y-R lymphoma cells. To our knowledge, this is the first report of antitumor activity of microalgae isolated from this geographical area.

Materials and Methods

Reagents, culture media, and tumor cells

L-glutamine and penicillin-streptomycin solutions were purchased from Life Technologies (Grand Island, NY). Concanavalin A (Con A), RPMI 1640 medium, fetal bovine serum (FBS), sodium dodecyl sulfate (SDS), N, N-dimethylformamide (DMF), phosphate buffered saline (PBS), and 3-[4,5-dimethylthiazol-2-yl]-2,5-diphenyltetrazolium bromide (MTT) were obtained from Sigma-Aldrich (St. Louis, MO, USA). Vincristine was obtained from Vintec (Columbia, S.A. de C.V., Ciudad de México). Extraction buffer was prepared by dissolving 20% (wt/vol) SDS at 37 °C in a solution of 50% each DMF and demineralized water, and the pH was adjusted to 4.7. The tumor cell line L5178Y-R (mouse DBA/2 lymphoma) was purchased from the American Type Culture Collection (LY-R, ATCC® CRL-1722™; ATCC, Rockville, MD, USA), maintained in culture flasks with RPMI 1640 medium supplemented with 10% FBS, 1% L-glutamine, and 0.5% penicillin-streptomycin solution (referred as complete RPMI medium) at 37 °C, in a humidified atmosphere of 5% CO2 in air. Cellular density was kept between 105 and 106 cells/mL.

Microalgae strains and culture

C. sorokiniana was isolated from San Juan River in the municipality of Cadereyta (26°21′55″N 98°51′15″O), whereas Scenedesmus was obtained from Pesquería River in the municipality of Apodaca (25°47′06″N 100°03′04″O) Nuevo Leon, Mexico. C. sorokiniana molecular identification using the P2F (5′-GGC TCA TTA AAT CAG TTA TAG-3′) and P2R (5′-CCT TGT TAC GA(C/T) TTC TCC TTC-3′) primers (Lee & Hur, 2009), which amplifies for a 1,700 bp fragment of the 18S gene, as previously reported by Cantú-Bernal (2017). Amplification conditions were an initial denaturation cycle at 95 °C for 5 min, 30–35 denaturation cycles at 95 °C for 30 s, alignment at 50–55°C for 30 s, and an extension process at 72 °C for 105 s, followed by a final extension at 72 °C for 7 min. The PCR product was confirmed by electrophoresis on 1.5% agarose gel at 100 Volts for 35 min, where the expected 1,700 bp band was observed. Once the PCR product was confirmed, the band was purified, for which the Wizard SV Gel and PCR clean-up system kit (Promega, Invitrogen) was used. For the band sequencing, the product was sent to the synthesis and sequencing unit of the Institute of Biotechnology, Universidad Nacional Autónoma de México. The edition and analysis of the Chlorella sp. sequence similarity percentage was carried out using the program Bioedit Sequence Alignment Editor v. 7.1.9 by sequence identity matrix means, after being compared with sequences reported in GenBank.

For microalgae culture, water samples were taken on 50 mL sterile Falcon tubes and kept at 5 °C ± 2 °C on ice. Then, 5 mL were transferred to 250 mL Erlenmeyer flasks, containing 100 mL of LC culture medium, as developed and reported by López-Chuken, Young & Guzman-Mar (2010). Flasks were then incubated at room temperature (25 °C ± 3 °C) in a continuous shaker at 120× g and under light radiation using 100 Watt white fluorescent light bulb as a continuous artificial light source (1,000 lux approximately). Flasks were incubated for 14 d until green growth was observed, after which, 100 µL were transferred to Petri dishes containing the same culture medium, but solidified with 1.5% of bacteriological agar. Inoculated dishes were incubated at 30 °C ± 2 °C by using a 100 watt white fluorescent light bulb as a continuous artificial light until isolated green colonies were observed. Single colonies were collected using a bacteriological loop and placed in Erlenmeyer flasks containing 100 mL of algal LC liquid culture medium. Next, flasks were incubated under the same conditions described above. This process allowed us selecting a single microalgae genus by picking up a single colony; however, given that microalgae tend to grow in consortia with bacteria and yeasts, microalgal cultures were treated with an antibiotic and antimycotic solution containing 500 UI/mL penicillin, 500 µg/mL streptomycin, 50 µg/mL gentamicin, and 1.25 µg/mL fungizone. For this, 5 mL of LC liquid culture medium with antibiotics were placed in 15 mL conical tubes, after which 0.25 mL of the algal culture were added and tubes were incubated for 48 h, under same shaking and lighting conditions described above. After the incubation period, 500 µL of the cultures were transferred into 50 mL of sterilized LC liquid culture medium without antibiotics producing axenic cultures of C. sorokiniana and Scenedesmus sp. isolates. Each axenic culture was grown for 14 d in 1-L Erlenmeyer flasks containing 500 mL of LC liquid culture medium, until exponential growth phase was reached (based on growth curve, Fig. S1). Next, each complete culture was transferred to individual bioreactor tanks containing 14.5 L of LC culture medium. Photobioreactor tanks were designed by the López-Chuken work team, and consisted of circular acrylic tanks of 30 cm of diameter and height; aeration was supplemented by air pumps with an adapted 0.2 µm filter at 1-L/min flow rate, radiated by continuous artificial LED white lights at 1,500 lux of intensity, and agitation by rotary plastic pallets at 50 rpm (Fig. S2). Biomass production in bioreactors was monitored every 2 d (Tuesdays, Thursdays, and Saturdays) by taking a 10 mL sample with a sterile pipette and filtering through a previously weighed 0.7 µm-pore size microfiber paper. Then the paper was dried at 70 °C inside an oven and weighed again; this monitoring process was repeated until the biomass production showed no increase. Once the maximum biomass production was reached, bioreactor tanks were stored at 4 °C ± 2 °C, until most microalgae biomass precipitated, then, the supernatant was decanted (Fig. S3). The collected wet biomass was the centrifuged at 9,000 rpm for 10 min (ST16R model; Thermo Fisher Scientific, Waltham MA, USA) and frozen dried (Labconco, Kansas City, MO, USA).

Biomass dried samples of C. sorokiniana and Scenedesmus sp. were placed in separate Whatman cellulose extraction thimbles (33 × 80 mm, thickness 1.5 mm) (Sigma-Aldrich, St. Louis, MO, USA) and placed in a Soxhlet extraction apparatus (Reyna-Martínez et al., 2015), which is a continuous system consisting of a flat bottomed round flask, an extraction chamber with a siphon, and a condenser. This method was selected since this extraction is very practical and recommended by most of the methanol-soluble compounds for biological material recovering. A round flask filled with 600 mL of methanol was used and the extraction lasted 48 h for each microalgae. Methanol was selected based on preliminary results where methanol extracts showed the highest cytotoxic activity against L5178Y-R cell line; whereas chloroformic extracts did not show cytotoxic effects and hexane itself showed cytotoxicity against the tumor cell line tested. After the biological material compounds were extracted with methanol, the solutions were filtered using Whatman filter paper, and solvent was evaporated using a rotary evaporator, leaving approximately 10 to 15 mL of liquid material. Remaining solvent was further removed by a vacuum desiccator. Extracts were dissolved in RPMI medium at a concentration of 1 mg/mL and kept frozen until use. From this stock, serial 1:1 dilutions from 500 to 7.8 µg/mL were prepared.

The tumor cell line L5178Y-R (mouse DBA/2 lymphoma) was purchased from the American Type Culture Collection (LY-R, ATCC® CRL-1722™; ATCC, Rockville, MD, USA), maintained in culture flasks with RPMI 1640 medium supplemented with 10% FBS, 1% L-glutamine, and 0.5% penicillin-streptomycin solution (referred as complete RPMI medium) at 37 °C, in a humidified atmosphere of 5% CO2 in air. Cellular density was kept between 105 and 106 cells/mL.

Tumor cytotoxicity and apoptosis assays

To determine the cytotoxic effect of C. sorokiniana and Scenedesmus sp. methanol extracts against L5178Y-R tumor cells, cell cultures were collected and washed three times in RPMI medium, then suspended and adjusted to 5 × 104 cells/mL with complete RPMI medium. One hundred microliters of the cell suspensions were then added to flat-bottomed 96-well plates (Becton Dickinson, Cockeysville, MD, USA), containing 100 µL of complete RPMI, methanol microalgae extracts at various concentrations, vincristine (250 µg/mL) as positive control, and RPMI medium as negative control; all treatments were tested in triplicate. Microplates were incubated for 48 h at 37 °C with 5% CO2, then 15 µL of MTT were added (0.5 µg/mL, final concentration), and cultures were incubated for 3 additional hours. After this, supernatant was removed and 80 µL of DMSO were added to all wells. Optical densities, resulting from dissolved formazan crystals, were then read in a microplate reader (DTX 880 Multimode detector; Becton Dickinson, Schwechat, Austria) at 570 nm (Gomez-Flores et al., 2009). The percentage of cytotoxicity was calculated as follows: %Cytotoxicity=100−A570in extract-treated cells/A570in untreated cells×100.

Apoptosis induction by C. sorokiniana and Scenedesmus sp. methanol extracts against L5178Y-R cell line was evaluated in vitro by acridine orange and ethidium bromide staining. For this, 1 × 106 L5178Y-R tumor cells were placed in 24-well plates in the presence of 500 µg/mL methanol extracts, and incubated for 24 h. Then, 500 µL of RPMI, plus 1-µL of acridine orange and 100 µg/mL ethidium bromide (1:1 ratio) were added to the wells. Next, cultured cells were incubated for 5 min, washed with 1-mL PBS, and suspended in 100 µL of RPMI medium; after incubation period, 10 µL of cell suspension were placed between a slide and a coverslip for fluorescence microscope visualization (Inverted Tissue Culture Fluorescence Microscope Olympus IX-70, Representaciones y Distribuciones FAL, S.A. de C.V., Naucalpan, State of México, Mexico). Acridine orange stains viable cells and dead cells (green cells), whereas ethidium bromide only stains those cells that have lost the integrity of their membrane (orange cells). Therefore, viable cells appear in a uniform green tone, cells found in apoptosis appear in a spotty green or granular in the center due to the condensation of chromatin and fragmentation of the nucleus, whereas cells in necrosis appear in a uniform orange hue (Coligan et al., 1995).

In addition, apoptosis induction was evaluated by DNA degradation (Orozco-Flores et al., 2017). For this, cells were incubated for 48 h with C. sorokiniana and Scenedesmus sp. methanol extracts at 500 µg/mL, testing their respective negative (culture medium) and positive (20 µg/mL Actinomycin D) controls. After the incubation period, cells were collected and centrifuged at 2,000 rpm for 10 min, then washed with PBS and extracted using the AxyPrep Multisource Genomic DNA Miniprep Kit (Axygen, Tewksbury, MA, USA). In order to visualize the extracted DNA, the sample was separated by 2% agarose gel electrophoresis, using SB buffer for the electrophoretic shift at 70 V for 20 min and 110 V for 1 h. After this, gel was stained with 5 ng/mL ethidium bromide and photographs were documented under High Performance Ultraviolet Transilluminator (UVP, LLC, Upland, CA, USA) light. DNA like-ladder fragmentation indicates apoptotic activity, whereas DNA smear represents cell death by necrosis.

In early apoptosis stages caspase enzymes are activated. Caspase participate in the cleavage of protein substrates leading to cell disassembly. Cleavage of protein substrates leads to a fluorescent monoamide formation and finally to a rhodamine 110 conversion. For apoptotic pathway involving caspases, caspase can be monitored by measuring fluorescence intensity using microplate wells (Towhid et al., 2013). For this, L5178Y-R cells (5 × 105 cells/well) were seeded in a 48 wells plate, and treated with Actinomycin D (800 ng/mL) as positive control, or Chlorella and Scenedesmus methanolic extracts at 500 µg/mL. Cultures were then incubated for 24 h at 37 °C, after which, activated caspases were detected with the CaspGLOW™ red active caspase staining kit following manufacturer’s instructions. Fluorescence intensity was measured at Ex/EM = 540/570 nm in a Varioskan Lux Multimode Reader (Thermo Fisher Scientific, Waltham, MA, USA).

Animals

Six- to eight-week old Balb/c female mice were purchased from Harlan Mexico S.A. de C.V. (D.F., Mexico). They were kept in a pathogen- and stress-free environment at 24 °C, under a light-dark cycle (light phase, 06:00–18:00 h) in a One Cage 2100™ System (Lab Products, Inc., Seaford, DE, USA), and given water and food ad libitum. Animals were euthanized by asphyxiation in a 100% CO2 chamber. Experiments involving the use of animals were reviewed and approved by our institutional animal care and use committee before being initiated, and were performed in accordance with the Guiding Principles in the Use of Animals in Toxicology, adopted by the Society of Toxicology in March 1999.

Murine thymus lymphocyte viability assay

Thymus was immediately removed after mouse death. Single-cell suspensions were prepared by disrupting the organ in RPMI 1640 medium. Cell suspensions were washed three times in this medium, and suspended and adjusted at 1 × 107 cells/mL in complete RPMI medium. Thymus lymphocyte viability was determined by a colorimetric technique using MTT (Gomez-Flores et al., 2009). Thymus suspensions (100 µL of 1 × 107 cells/mL) were added to flat-bottomed 96-well plates (Becton Dickinson, Cockeysville, MD, USA) containing triplicate cultures (100 µL/well) of complete RPMI medium (unstimulated control), or 100 µL of C. sorokiniana and Scenedesmus sp. methanol extracts at various concentrations, for 48 h at 37 °C in 95% air-5% CO2 atmosphere. After incubation for 44 h, MTT (0.5 mg/mL final concentration) was added, and cultures were additionally incubated for 4 h. Cell cultures were then incubated for 16 h with extraction buffer (100 µL/well), and optical densities, resulting from dissolved formazan crystals, were then read in a microplate reader (Becton Dickinson, Cockeysville, MD, USA) at 570 nm (Gomez-Flores et al., 2009).

All experiments were repeated at least three times with similar results. The results were expressed as means ± SEM of triplicate determinations from a representative experiment. Statistical significance was assessed by one-way analysis of variance and by the Student’s t test.

Results

Microscopic evaluation revealed the presence of C. sorokiniana and Scenedesmus sp. (Table 1), whose isolated colony cultures were then produced under axenic conditions. Culture of both microalgae in photobioreactor tanks showed that the exponential growth phase started after 12 d by C. sorokiniana, whereas for Scenedesmus sp., that started after 19 d (Fig. 1). Once the biomass was dried, collected, and weighed separately for each bioreactor, the yield by C. sorokiniana was of 0.24 g/L (±0.01), whereas for Scenedesmus sp. was of 0.30 (±0.01) g/L.

Figure 1 Biomass production time-course by Chlorella sorokiniana and Scenedesmus sp. isolates.

Data represent means ± SEM.

Table 1 Locations from Nuevo Leon state, Mexico, where microalgae were isolated.

Location	Microalga	Microscopic shape (100×)	
San Juan River, Cadereyta, N.L.
25°31′17″–100°0′34″	Chlorella sorokiniana		
Pesquería River, Apodaca, N.L.
25°46′34″–100°12′35″	Scenedesmus sp.		

Microalgae methanol extracts tested in vitro against tumor cell resulted in concentration-dependent activity against the murine tumor cell line L5178Y-R. C. sorokiniana extract caused significant (p < 0.05) 17% and 61% tumor cell toxicity at concentrations of 250 and 500 µg/mL, respectively, whereas that of Scenedesmus sp. induced 15%, 25%, and 75% cytotoxicity at concentrations of 125, 250, and 500 µg/mL, respectively (Fig. 2). Collected data were used to determine the inhibitory concentration mean (IC50) of C. sorokiniana and Scenedesmus sp., methanolic extracts. The observed IC50 for C. sorokiniana, Scenedesmus sp., and vincristine were 460.0 ±21.5, 362.9 ± 13.5, and 76.83 ± 2.55 µg/mL, respectively.

Figure 2 L5178Y-R tumor cell toxicity of Chlorella sorokiniana and Scenedesmus sp. methanol extracts.

Microalgae methanol extracts at concentrations ranging from 7.8 to 500 µg/mL were tested against L5178Y-R cells in vitro, as detailed in the text. Positive control vincristine caused 85% ±1.22 cytotoxicity at 250 µg/mL. Data represent means ± SEM. *P < 0.05; **P < 0.01.

C. sorokiniana and Scenedesmus sp. methanol extracts were shown to cause DNA fragmentation in L5178Y-R cells, with the typical ladder pattern, after 24 h of treatment, which was comparable with the results obtained with actinomycin D (Fig. 3A). Caspase activity assay showed that Scenedesmus sp. resulted in significantly higher (P < 0.05) apoptosis compared with the control (Fig. 3B). The AOPI staining analysis revealed that C. sorokiniana extract resulted in 74.4% tumor cell toxicity, 66% apoptosis, and 9% necrosis, whereas Scenedesmus sp. extract caused 54% tumor cell toxicity, 51% apoptosis, and 3% necrosis (Figs. 3B and 3D).

Figure 3 Apoptosis of L5178Y-R tumor cells.

(A) Agarose gel showing the cellular DNA fragmentation by cell line L5178Y-R after treatment with microalgae methanolic extracts. Lane 1, 100 bp molecular weight marker; lanes 2 and 3, cellular DNA after treatment with Chlorella sorokiniana methanolic extracts at 500 and 250 µg/mL, respectively; 4 and 5 cellular DNA after treatment with Scenedesmus sp. methanolic extracts at 500 and 250 µg/mL, respectively; lane 6, cellular DNA treated with actinomycin D at 20 µg/mL. (B) Detection of caspase 3/7 enzymes activity in L5178Y-R cells (CaspGLOW™), testing 500,000 cells per well on the same day, untreated or treated with Actinomycin D (800 ng/mL) as positive control, and C. sorokiniana or Scenedesmus sp. methanol extracts at 500 µg/mL, incubated by 24 h at 37 °C and reading fluorescence intensity at Ex/EM = 540/570 nm. (C) Effects of C. sorokiniana and Scenedesmus sp. methanol extracts on percent viable, apoptotic, and necrotic cells. Percentage of viable, apoptotic, and necrotic L5178Y-R cells after 24 h treatment with 500 µg/mL C. sorokiniana and Scenedesmus sp. methanol extracts, and actinomycin D (20 µg/mL). (D) L5178Y-R cells stained with acridineorange/ethidiumbromide used to discriminate viable, apoptotic and necrotic cells after C. sorokiniana and Scenedesmus sp. methanol extracts treatment.

Cytotoxicity of C. sorokiniana and Scenedesmus sp. methanol extracts did not significantly alter normal murine thymus lymphocyte viability, resulting in up to 26% and 19% toxicity with the highest extract concentration (500 µg/mL) (Fig. 4). In L5178Y-R tumor cells, 500 µg/mL of extract resulted in 62–75% toxicity (Fig. 2).

Figure 4 Thymus lymphocyte viability.

Effects of Chlorella sorokiniana and Scenedesmus sp. methanol extracts on viability of normal murine thymus lymphocytes. Thymus lymphocyte viability was determined by a colorimetric technique using MTT (Gomez-Flores et al., 2009). Thymus suspensions were incubated with culture medium alone or with C. sorokiniana and Scenedesmus sp. methanol extracts at various concentrations, for 48 h at 37 °C, and cell viability was determined as detailed in the text.

Discussion

To our knowledge, this is the first report of Nuevo Leon, Mexico native microalgae, identified as C. sorokiniana and Scenedesmus sp., showing cytotoxicity against a murine lymphoma tumor cell line. Previous reports have shown microalgae potential for wastewater treatment and biodiesel production (Reyna-Martínez et al., 2015; Beltrán-Rocha et al., 2017). In the present study, microalga isolates were grown under artificial light, closed photobioreactors, and previously established physicochemical conditions. Biomass production was monitored every 2 d by taking a 10 mL sample, filtered, dried, and weighed during the fermentation time-course. Biomass calculated from time-course data indicated that Scenedesmus sp. resulted in higher (twice as much) biomass production compared with that produced by C. sorokiniana, since values were higher than 0.8 and up to 0.4, respectively (Fig. 1). Nonetheless, after collecting the final biomass produced by each microalgae, Scenedesmus sp. resulted in only 18% more biomass, compared with C. sorokiniana, for a total of 13.74 g and 11.21 g dried biomass, respectively. C. sorokiniana and Scenedesmus sp. production in the photobioreactors was stopped after 29 d because no additional biomass production was observed. Biomass production was lower compared with other reports using batch culture and phototrophic conditions (Brennan & Owende, 2010; Chen et al., 2011). However, sufficient biomass was produced to obtain an adequate amount of methanol extract to perform biological assays.

In vitro tumor cell toxicity assays resulted in concentration-dependent activity against the murine tumor cell line L5178Y-R (up to 61.9% and 74.8% cytotoxicity at 500 µg/mL C. sorokiniana and Scenedesmus sp. extracts, respectively). These results are comparable with other reports showing about 50% in vitro cytotoxicity by microalga extracts against cervical cancer (Yusof et al., 2010; Kyadari et al., 2013).

Apoptosis is the best known pathway for programmed cell death. Apoptosis and necrosis can occur independently, sequentially or simultaneously. The type and/or the stimuli degree may determine if cells die by apoptosis or necrosis. At low doses, a variety of injurious stimuli such as heat, radiation, hypoxia, and cytotoxic anticancer drugs can induce apoptosis, or lead to necrosis at higher doses (Elmore, 2007). After cells enter the apoptotic process, their DNA degrades, showing a ladder pattern of multiples of approximately 200 base pairs, which can be observed when extracting the DNA and making an agarose gel electrophoresis. Apoptosis involves the activation of caspases enzymes linked to the initiating stimuli. Caspase-3 is required for apoptosis-associated chromatin margination, DNA fragmentation, and nuclear collapse of the cell (Mantena, Sharma & Katiyar, 2006). After testing C. sorokiniana and Scenedesmus sp. methanolic extracts, using the caspase-3/7 microplate assay, results demonstrated that only the Scenedesmus sp. methanolic extract was significantly different compared with the untreated cells (negative control), whereas no differences were observed with either actinomycin D or C. sorokiniana methanolic extract. Microalgae-induced tumor cytotoxicity was observed to be mediated by apoptosis, as determined by the acridine orange and ethidium bromide staining, as well as DNA fragmentation (ladder pattern) (Nagata, 2000). In fact, microalga isolates methanol extracts resulted in similar effects against the cell line compared with actinomycin D, compound that resulted in cellular apoptosis (Quintanilla-Licea et al., 2012). After testing crude extracts of the cyanobacteria Nostoc sp., against human pancreatic tumor cells PaTu 8902, Voráčová et al. (2017) found that apoptosis was mostly mediated by caspases 3 and 7. In summary, DNA fragmentation, acridine orange/ethidium bromide staining, and caspases results support apoptosis as the cell-death pathway by the tested microalgae methanolic extracts (Towhid et al., 2013).

Tumor cancer cells may develop as a result of in situ formation of nitrosamines from secondary amines and nitrite in an acidic environment of the stomach. There are chemical agents known as chemopreventers, which help to reverse, suppress or prevent these nitrosamines formation. In fact, ascorbic acid or phenolic compounds are chemopreventers, since they prevent or reduced nitrosamines formation (Jahan et al., 2017). It has been shown that microalgae synthesize a number of bioactive compounds, including bioactive peptides, fucans, galactans, alginates, phenolic compounds, phycocyanins, phycobiliproteins, eicosapentanoic and arachidonic acids, carotenoids, tocopherols, sterols, and terpenoids (Lordan, Ross & Stanton, 2011). Some of these compounds may be responsible for the cytotoxicity induced by C. sorokiniana and Scenedesmus sp. methanol extracts, against the murine lymphoma cell line L5178Y-R.

In a recent report, C. sorokiniana water extracts were evaluated against two human non-small cell lung cancer (A549 and CL1-5 human lung adenocarcinoma cells) cell lines using a subcutaneous xenograft tumor model. Results demonstrated the tumors growth inhibition after extract oral intake in vivo, through mitochondrial-mediated apoptosis (Lin et al., 2017).

In the present study, no significant lymphocyte cytotoxicity was observed by C. sorokiniana and Scenedesmus sp. methanol extracts. Nevertheless, results were comparable with other reports, which show low than 20% lymphocyte cytotoxicity but around 50% cytotoxicity against cervical cancer cells by microalga extracts in vitro (Yusof et al., 2010; Kyadari et al., 2013).

The bio-guided fractionation of these extracts is ongoing, and further studies of the isolated pure compounds will be performed.

Conclusion

The native microalgae C. sorokiniana and Scenedesmus sp. isolates from Nuevo Leon, Mexico water bodies were produced under a semi-pilot level using closed photobioreactors, with artificial illumination and aeration. The produced microalgae methanol extracts were cytotoxic against the murine tumor cell line L5178Y-R in vitro, by the mechanism of apoptosis, without affecting normal murine lymphocytes.

Supplemental Information

Figure S1 Timecourse of the microalgae culture in matraz

(A) Chlorella sorokiniana; (B) Scenedesmus sp.

Click here for additional data file.

Figure S2 Bioreactor tanks

Bioreactor tanks designed by López-Chuken, Young & Guzman-Mar (2010) consisting of circular acrylic tanks (30 cm of diameter and height) where aeration was supplemented by air pumps with an adapted 0.2 µm filter at 1-L min-1 flow rate; radiated by continuous artificial illumination (LED white lights at 1,500 lux of intensity), and under agitation by rotary plastic pallets at 50× g.

Click here for additional data file.

Figure S3 Microalgae production

Bioreactor tanks showing the Chlorella sorokinian a and Scenedesmus sp. for biomass production in tanks after 29 days fermentation.

Click here for additional data file.

Supplemental Information 1 IC50 of the Clorella sorokiniana methanolic extract

Click here for additional data file.

Supplemental Information 2 IC50 of the Senedesmus sp. methanolic extract

Click here for additional data file.

Supplemental Information 3 Caspases analysis

Click here for additional data file.

Supplemental Information 4 Raw data

Click here for additional data file.

Supplemental Information 5 Chloroformic and hexane extracts and vincristine cytotoxicity analysis dataset

Click here for additional data file.

Supplemental Information 6 Timecourse (14 days) of Chlorella y Scenedesmus culture in matraz (500 mL medio)

Click here for additional data file.

We thank Alonso A. Orozco-Flores and Enriqueta Monreal-Cuevas for technical assistance.

Additional Information and Declarations

Competing Interests

Author Contributions

Animal Ethics

Data Availability

The authors declare there are no competing interests.

Raul Reyna-Martinez conceived and designed the experiments, performed the experiments, analyzed the data, wrote the paper, prepared figures and/or tables, reviewed drafts of the paper.

Ricardo Gomez-Flores conceived and designed the experiments, analyzed the data, contributed reagents/materials/analysis tools, prepared figures and/or tables, reviewed drafts of the paper.

Ulrico López-Chuken conceived and designed the experiments, reviewed drafts of the paper.

Ramiro Quintanilla-Licea conceived and designed the experiments, analyzed the data, reviewed drafts of the paper.

Diana Caballero-Hernandez conceived and designed the experiments, performed the experiments, analyzed the data, reviewed drafts of the paper.

Cristina Rodríguez-Padilla contributed reagents/materials/analysis tools.

Julio Cesar Beltrán-Rocha performed the experiments, fermentor adaptation to improve photosyntesis by microalgae.

Patricia Tamez-Guerra conceived and designed the experiments, performed the experiments, analyzed the data, contributed reagents/materials/analysis tools, wrote the paper, prepared figures and/or tables, reviewed drafts of the paper.

The following information was supplied relating to ethical approvals (i.e., approving body and any reference numbers):

Balb/c female mice were purchased from Harlan Mexico S.A. de C.V. Experiments involving the use of animals were reviewed and approved by the Universidad Autonoma de Nuevo Leon animal care and use committee before being initiated.

The following information was supplied regarding data availability:

The raw data is included as a Supplemental File.

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
