# Peer review of "Antitumor activity of Chlorella sorokiniana and Scenedesmus sp. microalgae native of Nuevo León State, México"

_PeerJ, doi:10.7717/peerj.4358_

## Round 0.1 · original submission · Major Revisions

Both reviewers agree that more data are required to substantiate your claims:

A) representative images of cells stained with AO/EB and indicate apoptotic and necrotic examples
B) DNA laddering/smear figure is missing
C) Your paper would benefit greatly from the inclusion of additional apoptosis markers (annexin V, caspase).


Additionally, your results should be compared/contrasted with a recent paper showing that C. sorokiniana extract inhibits lung cancer in vitro and in vivo (BMC Complement Altern Med. 2017 Feb 1;17(1):88 DOI:10.1186/s12906-017-1611-9)

Please address also the other issues mentioned by our helpful reviewers

Reviewer 1 ·

Basic reporting

The introduction is too brief.

The results for DNA degradation in 2% agarose gel electrophoresis are not found in the text or Figure.

Experimental design

In the abstract the authors reported that
The extracts were shown to induce tumor cell death by the mechanism of apoptosis.
However, only one experiment was conducted (AOPI staining) to prove apoptosis (a cell death pathway). There were no experiments done to show the mechanism of apoptosis such as caspase activity and gene expression study.

Validity of the findings

Not enough data to prove the cytotoxic activity of the C. sorokiniana and Scenedesmus sp. methanol extracts. The results for DNA fragmentation is missing. Apoptosis should be supported by other assays such as cell cycle, Annexin 5, Mitochondrial membrane potential, caspase activity and apoptotic gene expression using qpcr.

The IC50 values of the extracts were not reported.

Additional comments

Not enough data to support the conclusion from the study. Hence, the results are inconclusive.
The experimental designs should be expanded to provide supporting results to justify the research question.

·

Basic reporting

1. English writing needs improvements.
2. In the discussion, need to cite and comment on recent paper showing that C. sorokiniana extract inhibits lung cancer in vitro and in vivo (BMC Complement Altern Med. 2017 Feb 1;17(1):88).
3. Should provide a representative image of cells stained with AO/EB and indicate apoptotic and necrotic examples in Fig. 3.
4. The methods mention DNA laddering/smear but this figure is not presented. This is very important to provide further confidence that the extracts are working through apoptosis (laddering) more than necrosis (smear), as suggested by AO/EB results in Fig. 3. In methods line 163, should state DNA smear not scan.

Experimental design

1. Appears to be original in that they obtained organisms from local river, However, there is one very recent but more complete paper mentioned above that uses C. sorokiniana extract in lung cancer.
2. Methods for isolation, growth, and preparation of extracts is well described. There are some issues in methods described in general comments.

Validity of the findings

1. AO/EB results would benefit from a representative image.
2. DNA laddering result is not shown and is essential before acceptance.

Additional comments

The authors obtained water samples from a local river in Nuevo Leon Mexico and isolated two strains of micoalgae, Chlorella sorokiniana and Scenedesmus. They present details on the methods of growing and confirming the identity of these organisms by 18S PCR. Methanol extracts (250-500 ug/ml) of dried mass obtained from bioreactors was used for testing anti-cancer efficacy in mouse lymphoma L5178Y-R cells. An acridine orange/ethidium bromide (AO/EB) staining assay was used to determine effects on apoptosis and necrosis. These extracts had less cytotoxic effects on non-cancer mouse thymocyes.

In general, this paper provides some interesting information on utilizing a local micoalgae organism for testing anti-cancer activity. However, there are several issues that need to be addressed.
1. Should provide a representative image of cells stained with AO/EB and indicate apoptotic and necrotic examples in Fig. 3.
2. The methods mention DNA laddering/smear but this figure is not presented. This is very important to provide further confidence that the extracts are working through apoptosis (laddering) more than necrosis (smear), as suggested by AO/EB results in Fig. 3. In methods line 163, should state DNA smear not scan.
3. In the discussion, need to cite and comment on recent paper showing that C. sorokiniana extract inhibits lung cancer in vitro and in vivo (BMC Complement Altern Med. 2017 Feb 1;17(1):88).
4. Sentence in lines 247-249 needs to be revised. It is assumed what is meant is that extracts have low cytotoxicity in non-cancer cells but higher in cancer.
5. English writing needs improvements.
6. Fig. 1, why after 18d Scenedesmus biomass much greater vs S. sorokiniana? Does not correspond to numbers given.
7. Several issues in methods section: a) more information of 18S PCR to identify strains; b) “Tumor cytotoxicity and apoptosis assay” subheading is misplaced and should be moved to line 120; c) how is the concentration of the extract measured? It is given as 1 mg/ml; d) more information on microscope source, wavelength used for analysis; e) there is mention of vincristine as positive anti-cancer drug but actinomycin D is used in Fig. 3.

---

## Round 0.2 · Minor Revisions

Thank you for the thorough responses to our reviewer's requests. Before I send the paper to a final round of revision, I will need you to address a few issues:

A) In this revision, you have inadvertently replaced your intended Fig.1 by Fig.2, and your intended Fig. 2 by a version of Fig.4 with different shadings of the bars.

B) Some language issues remain, a few of which are listed below:
i) line 54 The following sentence is not very clear, and it should probable be rephrased: "Furthermore, the use of polluting elements such as nitrogen, phosphorus, and sulphur for microalgae growth can be considered an advantage, since such elements can be also metabolized by harmful aquatic weeds to proliferate. "
ii) in line 167, you state "previous results ", which would seem to refer to previously published data. Do you mean "preliminary results" instead?
iii) line 361 "intake in vivo, being mitochondrial-mediated " should be "intake in vivo, through mitochondrial-mediated "
iv) line 363 " not significant" should be "no significant", and line 365 "reports, showing low" should be ""reports, which show low"

C) I am afraid that the discrepancy between the biomass data plotted in Fig.1 and the 18% difference in biomass between both algae remains unexplained and is probably unresolvable (since it can be due to a number of factors such as insufficient drying of Scenedesmus samples post-18d , unrepresentativeness of those samples due to aggregation or insufficient shaking/homogeneization, etc). This is obviously a very minor point in your paper.

---

## Round 0.3 · Minor Revisions

Unfortunately current figure 2 still depicts murine lymphocyte viability instead of tumor toxicity. Please correct.

---

## Round 0.4 · accepted · Accept

Thank you for addressing all the reviewer's concerns. There are minor comments in the 2 appended reviews which you should address while in Production.

Reviewer 1 ·

Basic reporting

No comment

Experimental design

No comment

Validity of the findings

No comment

Additional comments

Some modifications have been made in the revised manuscript. I have attached the pdf file of the edited manuscript.

Annotated reviews are not available for download in order to protect the identity of reviewers who chose to remain anonymous.

·

Basic reporting

1. Line 282--- misspelled ladder (“latter”).
2. Line 290-291---revise to read “…resulting in up to 26% and 19% toxicity with the highest extract concentration (500 g/ml) (Fig. 4). In L5178Y-R tumor cells, 500 g/ml of extract resulted in 62-75% toxicity (Fig. 2)."

Experimental design

Acceptable.

Validity of the findings

Acceptable.

Additional comments

Minor writing changes. Paper is improved and is now acceptable.